# Dickkopf-Related Protein 1 (DKK-1) as a Possible Link between Bone Erosions and Increased Carotid Intima-Media Thickness in Psoriatic Arthritis: An Ultrasound Study

**DOI:** 10.3390/ijms241914970

**Published:** 2023-10-07

**Authors:** Cristina-Elena Biță, Ștefan Cristian Dinescu, Anca-Lelia Riza, Paulina Lucia Ciurea, Anca Emanuela Mușetescu, Daniela Marinescu, Roxana Mihaela Dumitrașcu, Larisa Ionela Șuiu, Răzvan Adrian Ionescu, Horațiu Valeriu Popoviciu, Florentin Ananu Vreju

**Affiliations:** 1Department of Rheumatology, University of Medicine and Pharmacy of Craiova, 2 Petru Rareş Street, 200349 Craiova, Dolj County, Romania; cristina.gofita@umfcv.ro (C.-E.B.); paulina.ciurea@umfcv.ro (P.L.C.); anca.musetescu@umfcv.ro (A.E.M.); florentin.vreju@umfcv.ro (F.A.V.); 2Human Genomics Laboratory, University of Medicine and Pharmacy of Craiova, 2 Petru Rareş Street, 200349 Craiova, Dolj County, Romania; 3Regional Center for Medical Genetics Dolj, Emergency County Hospital Craiova, 1 Tabaci Street, 200642 Craiova, Dolj County, Romania; 4Department of General Surgery, University of Medicine and Pharmacy of Craiova, 2 Petru Rareş Street, 200349 Craiova, Dolj County, Romania; daniela.marinescu@umfcv.ro; 5Doctoral School, University of Medicine and Pharmacy of Craiova, 2 Petru Rareş Street, 200349 Craiova, Dolj County, Romania; roxana.dumitrascu21@gmail.com (R.M.D.); larisasuiu10@yahoo.com (L.I.Ș.); 6Third Internal Medicine Department, ‘Carol Davila’ University of Medicine and Pharmacy, 8 Eroii Sanitari Avenue, 050471 Bucharest, Romania; tane67@gmail.com; 7Department of Rheumatology, BFK and Medical Rehabilitation, University of Medicine, Pharmacy, Science and Technology of Targu Mures, 38 Gh. Marinescu Street, 540142 Târgu Mureș, Mureș County, Romania; phoratiu2000@gmail.com

**Keywords:** DKK-1, cardiovascular risk, atherosclerosis, erosions, psoriatic arthritis

## Abstract

Psoriatic arthritis (PsA) is a heterogenous systemic inflammatory disorder that affects peripheral joints and skin, but also causes inflammation at entheseal sites, digits (dactylitis) and the axial skeleton. Despite considerable advances, our understanding of the pathogenesis and management of PsA is hampered by its complex clinical expression. We enrolled patients who met the ClASsification for Psoriatic Arthritis (CASPAR) criteria for PsA (*n* = 17), and healthy controls (*n* = 13). The lipid profile, C-reactive protein (CRP) and Dickkopf-related protein 1 (DKK-1) circulating levels were measured for all subjects. For the patients with PsA, (1) the erosive character of the articular disease was assessed by a musculoskeletal ultrasound and (2) the cardiovascular risk was evaluated using the Systematic Coronary Risk Evaluation (SCORE) chart and the ultrasound measurement of the carotid intima-media thickness. A higher titer of serum DKK-1 was associated with the presence of erosions (*p* < 0.005) and the cIMT correlated with DKK-1 levels in patients with PsA (r = 0.6356, *p* = 0.0061). Additionally, we observed a positive correlation between increased cIMT and CRP (r = 0.5186, *p* = 0.0329). Our results suggest that DKK-1 could be used as an early biomarker for the erosive character of the articular disease and for the assessment of the cardiovascular risk in PsA patients.

## 1. Introduction

Psoriatic arthritis (PsA) is a chronic, multifaceted inflammatory condition of the joints that impacts approximately one-third of individuals diagnosed with psoriasis [1]. PsA usually occurs among individuals whose age ranges from 30 to 50 years, and arises from a complex synergy of genetic predisposition, local factors specific to the affected areas (skin, joints and entheses/spine) and environmental stimuli, such as biomechanical stress. These factors interact with the innate and adaptive immune system, ultimately shaping the observed clinical phenotypes. It impacts both peripheral joints and the spine, leading to irreversible damage and heightened functional impairment through bone erosions and aberrant bone growth.

To achieve a comprehensive understanding of PsA, one must consider the diverse manifestations of dynamic bone remodeling. These include large eccentric erosions, tuft resorption, periostitis, ankylosis and mutilans deformities. Notably, PsA patients often exhibit diffuse bone marrow and perientheseal edema, which are less frequently observed in rheumatoid arthritis (RA). Peripheral joints in PsA may also exhibit osteitis, a crucial characteristic of axial disease in radiographic axial spondyloarthritis, formerly known as ankylosing spondylitis (AS). Peripheral osteitis potentially contributes to the significant bone remodeling seen in patients with focal eccentric erosions and arthritis mutilans [2].

In recent studies, soluble mediators involved in bone remodeling were identified in the bloodstream of PsA patients. For example, Dickkopf-related protein (DKK-1), which is induced by tumor necrosis factor (TNF), hinders the function of osteoblasts and is found to be elevated in RA, potentially explaining the impaired joint repair mechanism. Interestingly, individuals with AS exhibit low levels of serum DKK-1, which could contribute to the characteristic osteoproliferation observed in this condition [3]. However, the precise level of DKK-1 in PsA is currently uncertain.

In addition to musculoskeletal involvement and extra-articular manifestations, such as dermatological, digestive or ophthalmological, patients with PsA also have multiple cardiovascular comorbidities [4]. Cardiovascular disease (CVD) is often accompanied by an elevated occurrence of the metabolic syndrome and its primary characteristics, such as hypertension, hyperlipidemia, obesity and impaired fasting glucose, and thus, represents a leading cause of mortality in PsA. The immune-mediated inflammatory mechanisms that underlie PsA may play a critical role in endothelial dysfunction, atherosclerosis (ASc) and the early onset of CVD [5,6].

The research findings additionally suggest that the Wnt signaling pathways play a significant role in modulating inflammation. These pathways are regulated by various antagonists, such as DKK-1. Studies have demonstrated that DKK-1, originating from platelets and endothelial cells, has the ability to stimulate leukocyte migration, activate endothelial cells and promote inflammation [7]. Such processes can potentially contribute to the progression of ASc.

Clinical studies have revealed elevated concentrations of DKK-1 in ASc, particularly in cases of advanced and unstable disease, as well as in acute ischemic stroke [8]. As a result of DKK-1’s functional role in regulating inflammation mediated by platelets and its contribution to plaque destabilization, it can be hypothesized that DKK-1 could be used as a risk predictor and as a potential target for future therapies in ASc.

The assessment of carotid intima-media thickness (cIMT) measured by Doppler ultrasonography is a marker for early, generalized ASc, while the presence of atheroma plaques indicates advanced ASc. Carotid intima-media thickness is a valuable noninvasive measure of macrovascular ASc, which can be assessed using high-resolution B-mode ultrasound [9,10]. Carotid ultrasound examination results lead to the reclassification of a significant proportion of patients within risk categories based on the SCORE (Systematic Coronary Risk Evaluation) charts and provides early information about the presence of ASc in its subclinical stages among individuals at risk.

Thus, the aim of the study is to assess if DKK-1 plays a predictive role in the increased cardiovascular risk in patients with PsA. We evaluated the association between PsA activity, articular and cardiovascular damage, metabolic changes and serum DKK-1 levels.

## 2. Results

### 2.1. Study Group Characteristics

A total of 17 patients were included in the study, consisting of 9 women and 8 men. The participants had a mean age of 49.53 ± 9.65 years, and the average duration of the disease was 5.88 ± 4.28 years, ranging from 1 to 16 years. Upon analyzing the groups under study in terms of the lipid profile, we observed elevated levels of TC and LDL, as well as a decreased level of HDL in the patient group. However, these differences did not reach statistical significance. We registered a mean body mass index (BMI) of 28.89 ± 5.47, with four patients (23.53%) being overweight and seven (41.18%) obese. Table 1 provides an overview of the general characteristics of the study group.

Concerning the inflammatory markers, we observed an average CRP value of 16.9 ± 12.47 mg/dL. Disease activity scoring revealed an average DAPSA score of 36.76 ± 12.26. The PASI score showed a mean value of 17.41 ± 11.12, indicating significant psoriasis severity. Enthesitis, as defined by OMERACT, was present in 11 of the participants (64.71%), with the Achilles tendon being the most commonly affected site of inflammation among our patients, in accordance with other published studies [11]. Analyzing our data to assess the impact of BMI on entheseal damage, we observed a Pearson’s correlation coefficient of 0.8394. Thus, it is worth mentioning that the ultrasound examination revealed a substantial occurrence of enthesitis in patients who were classified as overweight or obese. Additionally, the structural, irreversible joint damage was highlighted by bone erosions, with 52.94% of the patients presenting an erosive disease (Figure 1, Table 2). Due to the exclusive use of csDMARDs in the patients included in the study, achieving a tight control of disease activity was not attained. Consequently, the ultrasonography evaluation revealed an active Power Doppler (PD) enthesitis rate of 47.05% in the assessed sites, with the Achilles tendon (AT) being the most commonly affected area, accounting for 29.41% of the cases. Since the musculoskeletal ultrasound assessment was performed by three evaluators, our study also included the interrater reliability and found a good reproducibility, with a value of the ICC of 0.773.

Moreover, in agreement with the SCORE risk charts, the selected patients had a risk of 13.47% of developing a fatal or non-fatal CVD event in ten years. Additionally, the mean value of cIMT was 0.68 ± 0.09 mm, and 52.94% of the individuals showed the presence of atherosclerotic plaques in their carotid ultrasound scans (Figure 2). A limitation of our study was the fact that the measurement of the cIMT was performed by only one vascular ultrasonographist.

### 2.2. Correlations of DKK-1 and cIMT for the Evaluation of CVD

A Pearson’s correlation analysis was conducted to investigate the relationship between DKK-1 and other factors associated with the development of CVD. In patients with PsA, a positive correlation was found between cIMT and DKK-1 levels (r = 0.6356, *p* = 0.0061). The average cIMT (0.68 mm) observed in this study is consistent with the findings from previous reports on PsA patients [12,13]. This finding is particularly noteworthy considering that the median age of our patient population was only 49 years, a group traditionally assumed to have a lower burden of atherosclerotic CVD compared to those in higher age groups. Furthermore, a DKK-1 ROC curve was performed to assess the overall diagnostic performance in atherosclerosis. We observed a similarity between the DKK-1 (0.7466, *p* = 0.0226) and cIMT (0.7647, *p* = 0.0144) area under the ROC curve (AUC), while the optimum cutoff value for DKK-1 was 8482 pg/mL (sensitivity 82.35%; specificity 61.54%) and for cIMT was 0.6 mm (sensitivity 76.47%; specificity 84.62%). The data were also statistically significantly larger in comparison to AUC = 0.5, borderline of the diagnostic usefulness of a test (Figure 3).

Furthermore, we observed a significant correlation between increased cIMT and CRP (r = 0.5186, *p* = 0.0329), indicating that systemic inflammation in PsA could potentially accelerate the progression of ASc [9,14]. CRP, a widely recognized surrogate marker of systemic inflammation, is known to trigger endothelial cell activation and promote atherosclerotic processes [15,16]. Consistent with previous findings, our study revealed that elevated levels of CRP (16.9 mg/dL) may contribute to the accelerated development of ASc in PsA.

### 2.3. Correlations of DKK-1 and Bone Erosion

Together with the correlation between DKK-1 titers and cIMT, our results confirm that a higher titer of serum DKK-1 is associated with the presence of erosions. The mean value of DKK-1 for patients without erosions was 8046 pg/mL, while for patients with erosions, it was significantly higher (13,600 pg/mL; *p* = 0.0016). Taking into consideration the above-mentioned facts, DKK-1 could be used as an early biomarker for the erosive character of the articular disease and for the assessment of the cardiovascular risk (CVR) in PsA patients.

Therefore, as DKK-1 is highly associated with both CVD and erosions, one might reasonably link cIMT and bone erosions, pointing out the importance of the complex evaluation in patients with PsA. The mean value of the cIMT for patients without erosions was 0.57 mm, while for patients with erosions, it was 0.78 mm (*p* = 0.0052; Figure 4).

## 3. Discussion

PsA is a chronic inflammatory condition linked to numerous non-joint-related characteristics and associated health conditions. These factors indicate the presence of a systemic state of inflammation, endothelial dysfunction and atherosclerosis, which can occur even during the early phases of the disease before noticeable clinical signs and symptoms emerge [17,18]. Consequently, the term “psoriatic disease” (PsD) has gained prominence in recent years, emphasizing the concept that shared inflammatory and metabolic pathways are activated in various cells and tissues, including adipose tissue [19], synovium and skin [20] as well as endothelial cells [21].

Multiple studies have confirmed and validated the presence of metabolic syndrome in individuals with PsA or psoriasis, and our own study is aligned with these findings by observing similar results. Specifically, an altered lipid metabolism is frequently observed in these patients, with high titers of TC and LDL levels being the most commonly reported abnormalities. Additionally, we observed lower levels of HDL (Table 1), a well-established cardiovascular risk factor, which is consistent with the findings from several other publications [22,23].

The advent of targeted biological therapy has ushered in significant progress in the management of psoriasis and PsA, enabling a subset of patients to achieve remission. Treatments, such as anti-TNF therapy, interleukin-12/23-targeted therapy and interleukin-17 blockade, have shown effectiveness in cases of previous treatment-resistant disease [24,25]. However, a small percentage of patients either encounter severe side effects or do not respond to these interventions. In addition to assessing disease activity, the use of biomarkers has the potential to establish an early diagnosis and guide treatment decisions.

Our comprehension of the molecular mechanisms through which DKK proteins control both Wnt signaling and Wnt-independent pathways has significantly advanced. This knowledge has revealed that the abnormal regulation of DKK protein function plays a significant role in the development of various diseases, primarily cancer, neurodegenerative conditions and bone disorders [26,27].

The Wnt signaling pathway is an intricate system that encompasses a wide range of ligands, receptors and coreceptors, and its control takes place through multiple mechanisms. Among the regulators of Wnt signaling, the DKK protein family has been extensively studied and characterized [28]. More recently, the role of DKKs in cardiovascular research has gained attention, and emerging evidence suggests their involvement in the pathophysiology of the arterial wall, particularly in the regulation of atherosclerosis.

Our study indicates that DKK-1 may serve as an independent risk predictor for ASc, potentially reflecting its role in the underlying pathophysiology. In our investigation involving patients with PsA, we observed a relatively high prevalence of ASc. Additionally, our findings reveal that patients with PsA had a 13.47% ten-year risk of experiencing a fatal or non-fatal cardiovascular event. Moreover, we confirmed a positive correlation (r = 0.3985) between cIMT and cardiovascular risk as defined by SCORE. Although the statistical significance of this result is limited (*p* = 0.1131), increasing the sample size could enhance its reliability. However, multiple studies from the past decade have indicated that conventional tools, such as SCORE or Framingham risk charts, might not accurately estimate the actual cardiovascular risk in individuals with PsA and other inflammatory rheumatic conditions [29,30,31]. The decreased sensitivity of these estimators in assessing cardiovascular risk might be attributed, at least in part, to the exclusion of inflammation markers from their risk algorithms. This hypothesis is supported by evidence showing that the suppression of inflammation through anti-TNF agents has a beneficial effect on ASc and arterial stiffness in PsA patients [32].

Our findings reveal a noteworthy positive correlation between DKK-1 levels and cIMT, which serves as an indicative measure of early-stage atherosclerosis. The cIMT includes the combined thickness of the endothelium, connective tissue and smooth muscle. It is at this precise location that lipid deposition and plaque formation take place, underscoring the significance of cIMT as a marker for assessing atherosclerotic alterations in blood vessels.

Enthesitis, characterized by inflammation at the site where tendons or ligaments attach to the bone, is a prominent feature of PsA and is more commonly observed in PsA patients compared to individuals with other inflammatory or non-inflammatory diseases. Enthesitis is directly linked to both peripheral and axial structural damage. Identifying enthesitis through clinical examination alone can be difficult, especially for asymptomatic patients or those with signs and symptoms resembling other conditions. Thus, additional imaging techniques are often required. US is a well-established and validated approach for detecting peripheral enthesitis [33], enabling the identification of both subtle and clinically overt manifestations in patients with PsA. It offers precise information regarding structural changes and the presence of inflammation.

The detection of perientheseal edema through magnetic resonance imaging (MRI) and the identification of subclinical enthesitis using ultrasound in PsA emphasize the involvement of the enthesis in the pathogenesis of PsA [34]. In order to elucidate the imaging findings observed, the notion of an “entheseal organ” has been proposed. This theoretical model suggests that the enthesis, along with the adjacent bursa and synovium, form an integrated functional unit. Factors specific to this tissue, triggered by altered biomechanics within a particular genetic context, initiate an innate immune response that extends to the neighboring synovial and bursal tissues [35]. Regarding the prevalence of entheseal involvement in PsA, our ultrasound examination revealed the presence of enthesitis in 67.71% of the patients.

In addition, structural damage is one of the well-established factors associated with increased mortality in PsA [36]. The process of bone remodeling involves the coordinated actions of osteocytes, osteoblasts and osteoclasts, with regulation influenced by locally produced signaling molecules, hormones and cytokines, all of which can be influenced by mechanical stimuli. Therefore, it is worth considering that body weight and metabolic conditions may act as confounding factors for enthesopathic structural lesions. We noted a positive correlation between enthesitis and BMI, with a higher prevalence of entheseal abnormalities observed in overweight and obese patients. BMI, a variable of notable significance in determining entheseal findings, tends to be elevated in PsA patients compared to individuals without the condition.

The maintenance of bone homeostasis is believed to rely on the Wnt signaling pathway. DKK-1, an essential regulator of the Wnt pathway, acts as an inhibitor of osteoblast activity. However, there is compelling evidence suggesting that TNF can play both beneficial and detrimental roles in osteoblastogenesis, and these roles may vary depending on the differentiation stage of the cells [37].

Furthermore, osteocytes release DKK-1 in response to TNF, resulting in the inhibition of the Wnt pathway and subsequent impairment of new bone formation. Additionally, compelling evidence supports the notion that the elevated expression of DKK-1 contributes to the development of erosive bone disorders, such as PsA [38]. These findings, along with studies involving animal models of other inflammatory arthritic conditions, clearly demonstrate the dual regulatory role of DKK-1 in both osteoblast and osteoclast functions.

The soluble biomarker examined in this study plays a crucial role in regulating bone turnover, and our findings indicate that the systemic expression of factors that promote osteoclastogenesis and bone loss, such as DKK-1, is dysregulated in patients with PsA. Regarding structural changes, our study established a statistically significant association between serum DKK-1 levels and the presence of joint erosions, as detected in ultrasound examinations, which serve as indicators of bone damage. This finding aligns with the observations made by other studies [39], who reported that PsA patients with elevated DKK-1 levels presented a significantly higher occurrence of bone erosions compared to those with normal DKK-1 levels. However, there are contradictory findings in the literature regarding DKK-1 expression in PsA. Fassio et al. [40] observed that PsA patients had lower levels of DKK-1 compared to both RA patients and the healthy controls. Conversely, a study conducted in New Zealand reported that individuals with PsA exhibited higher serum concentrations of DKK-1 when compared to the healthy controls [3,39]. Thus, an additional aspect to take into account could be that serum DKK-1 levels may vary among phenotypes.

Nevertheless, our study shares the inherent limitations of cross-sectional studies. Therefore, determining the causal relationship of the identified positive associations through multivariate analysis becomes challenging. Moreover, due to the small sample size in our study, the predictive value of our research regarding the incidence of future adverse cardiovascular events remains uncertain. Thus, a future, more relevant study should include a larger number of subjects with a broader disease duration. Additionally, another limitation of the study was the inclusion of cases on a single occasion, which restricts our ability to assess longitudinal changes.

Considering that PsA is a disease with significant joint involvement, periodic evaluations are useful for assessing the frequently affected structures using ultrasonography. Therefore, performing musculoskeletal ultrasound allows for the identification of structural changes, such as enthesitis and erosions. In addition to joint involvement, PsA remains a disease associated with multiple cardiovascular comorbidities. Considering that both our data and other studies [41,42,43] have found a positive correlation between the presence of erosions and cIMT, which is a surrogate marker of CVD, evaluating joint involvement, even in the early stage of the disease, could help in stratifying the CVR in PsA patients. Thus, DKK-1 can be viewed as a common biomarker for both bone remodeling and the presence of early ASc. Therefore, measuring DKK-1 could be an important predictive marker for the aggressiveness of PsA, both in terms of erosive joint disease and increased cardiovascular risk.

## 4. Materials and Methods

### 4.1. The Study Cohort

A series of 17 consecutive patients who met the CASPAR criteria for PsA [44], admitted into the Emergency County Hospital of Craiova, Romania, Rheumatology Department, between April 2022 and April 2023, were assessed. Moreover, concerning the clinical patterns of PsA, we included patients exhibiting peripheral involvement, without any indications of axial disease based on radiographic evaluation. Nevertheless, it is important to note that our study did not directly compare the radiographic findings of the peripheral involvement in PsA with other forms of arthritis, such as RA or osteoarthritis. These limitations should be acknowledged and addressed in future studies. The exclusion criteria were: patients younger than 18 years, pregnant and lactating females, other forms of inflammatory arthritis, patients with impaired liver and kidney functions, diabetes mellitus, history of cardiovascular or cerebrovascular disease, thyroid dysfunction, multiple sclerosis, human immunodeficiency virus and alcohol-associated liver disease (Figure 5). The control group comprised 13 healthy subjects, who were carefully matched in terms of age and sex (Table 1). Moreover, they had no family history of psoriasis, RA, PsA or any other inflammatory rheumatic disease.

The study population and the control group were ethnically homogeneous, belonging to a mixed rural and urban Caucasian population. The research was conducted in adherence to the ethical and deontological principles outlined in the Helsinki Human Rights Declaration and received approval from the Ethics Committee of the Emergency County Hospital of Craiova. Prior to inclusion, written informed consent was signed by each participant.

All participants, including those in the healthy control group, underwent the collection of demographic data, recording the relevant medication usage and medical history, total cholesterol (TC), high-density lipoprotein cholesterol (HDL), low-density lipoprotein cholesterol (LDL), C-reactive protein (CRP) and serum analysis for various biomarkers, such as DKK-1 (soluble mediator of bone remodeling). In order to differentiate from other forms of arthritis, all patients underwent testing for rheumatoid factor and anti-cyclic citrullinated peptide antibody. Patients with PsA underwent additional assessments, including the evaluation of disease activity using the DAPSA (Disease Activity in Psoriatic Arthritis) score and musculoskeletal ultrasound. The severity of skin psoriasis in patients was evaluated using the Psoriasis Area and Severity Index (PASI).

Regarding the treatment, conventional synthetic disease-modifying antirheumatic drugs (DMARDs) were administered to 82.35% of the patients. Among them, 52.94% received methotrexate, 23.53% received leflunomide and 5.88% received sulfasalazine. It is worth noting that none of the selected patients had prior exposure to biological DMARDs or targeted synthetic DMARDs.

### 4.2. Laboratory Analysis

Peripheral venous blood samples were collected between 8 and 10 am. Sample processing, biobanking and biomarker analysis were performed at the Laboratory of Human Genomics, University of Medicine and Pharmacy, Craiova. Following a 15 min, 3000 RPM centrifugation to separate the serum, the samples were stored at −20 °C in Eppendorf tubes until analysis. The Human DKK-1 colorimetric sandwich Enzyme-Linked Immunosorbent Assay (ELISA) kit (Abcam, Waltham, MA, USA, ab100501) was used according to the manufacturer’s instructions. We extrapolated the serum-concentrated DKK-1 in the serum by measuring the optical density (O.D.) at 450 nm and 630 nm in a micro-plate reader (BMG CLARIOstar). The sensitivity declared by the manufacturer was <100 pg/mL and the detectable range was 122.9–30,000 pg/mL.

### 4.3. Musculoskeletal Ultrasound

The ultrasonographic (US) assessment included patients with PsA, diagnosed by a rheumatologist. The assessment was conducted by three skilled sonographers who were unaware of the patient’s history, clinical findings and biological information. An Esaote MyLab X8 machine, equipped with a high-frequency linear probe (4–15 MHz), was used for the examination. The tendons were evaluated in two planes: longitudinal, parallel to the tendon fibers, and transverse, perpendicular to them. Each gray-scale examination was coupled with a Doppler examination, with the pulse repetition frequency (PRF) of 700 Hz for the analysis of the vascularization index of the entheses. Thus, both the color Doppler and power Doppler modes were used for better sensitivity. The tendons were viewed in tension for anatomical details in the gray scale, as well as in the neutral position for the Doppler evaluation.

The following anatomical structures were assessed: right/left upper limb tendons (common finger extensor, common finger flexor and triceps), right/left lower limb tendons (quadriceps, patellar, peroneal, posterior tibial, Achilles and plantar fascia). For each examined structure, the presence of acute or chronic lesions was recorded. Acute lesions (increase in tendon thickness, reduced echogenicity and increased vascularity index) or chronic lesions (irregularities of the bone cortex, presence of enthesophytes and intratendinous calcifications) were assessed.

Furthermore, the identification of the pathological aspects was conducted according to the Outcome Measures in Rheumatology (OMERACT) and the MAdrid Sonographic Enthesitis Index (MASEI) definitions of ultrasound lesions [11,45,46].

### 4.4. Cardiovascular Risk Estimation

#### 4.4.1. SCORE Chart

The likelihood of fatal atherosclerotic CVD events occurring within a 10-year timeframe was determined by employing the SCORE chart specifically calibrated for very-high-risk countries (such as Romania), in accordance with the European guidelines [47].

#### 4.4.2. Measurement of Intima-Media Thickness

The carotid intima-media thickness was measured using a high-resolution Doppler ultrasound Canon Aplio 400 (TUS-A400 model) with a linear 8–10 MHz transducer. The scans involved capturing B-mode images of the left and right common carotid arteries, the carotid bulb, external carotid artery and internal carotid artery. The average cIMT of the right and left common carotid arteries was assessed at least at 1 cm proximally from the origin of the bulb. An atherosclerotic plaque was identified as a localized thickening of the vessel wall that was at least 50% greater than the surrounding region or as a distinct intimal thickening protruding into the lumen, measuring more than 1.5 mm. All studies were conducted by a single vascular ultrasonographist who was blinded for the clinical data.

### 4.5. Statistical Analysis

GraphPad Prism 8 (GraphPad Software, Inc.) (San Diego, CA, USA) was employed for statistical analysis. The results are presented as mean ± SD, and the data were assessed for normality using the Kolmogorov–Smirnov test. Group comparisons were conducted using a *t*-test, while correlations were evaluated using the Pearson’s coefficient and the diagnostic performance with a ROC curve. The optimum cutoff values were obtained using the sum of squares method [48]. The interrater reliability of the musculoskeletal ultrasonography changes was assessed using the intraclass correlation coefficient (ICC). ICC values lower than 0.5, between 0.5 and 0.75, between 0.75 and 0.9, and greater than 0.90 are indicative of poor, moderate, good, and excellent reliability, respectively [49]. We considered a level of *p* < 0.05 to be statistically significant.

## 5. Conclusions

In addition to their role in screening for PsA, soluble biomarkers can also assess the likelihood of developing cardiovascular disease in individuals with a diagnosis of PsA. The levels of DKK-l in the bloodstream show promise as a prognostic biomarker for PsA activity and the extent of joint erosions and damage. Thus, maintaining a tight control over inflammation and striving to achieve specific treatment goals, such as minimal disease activity response, can effectively reduce the risk of structural damage and, consequently, minimize the elevated cardiovascular risk associated with this condition.

## Figures and Tables

**Figure 1 ijms-24-14970-f001:**
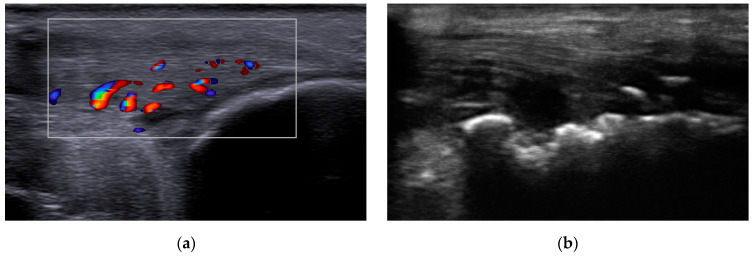
Musculoskeletal ultrasound evaluation: (**a**) Achilles tendon enthesitis showing increased tendon thickness and the presence of intratendinous Doppler signal; (**b**) erosions of the calcaneal bone at the insertion of the Achilles tendon.

**Figure 2 ijms-24-14970-f002:**
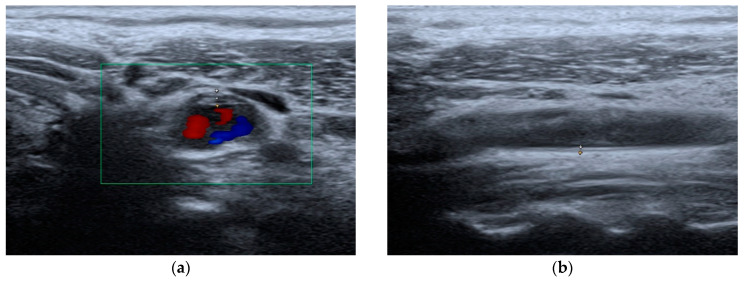
Carotid artery ultrasound evaluation: (**a**) atherosclerotic plaque; (**b**) cIMT measurement.

**Figure 3 ijms-24-14970-f003:**
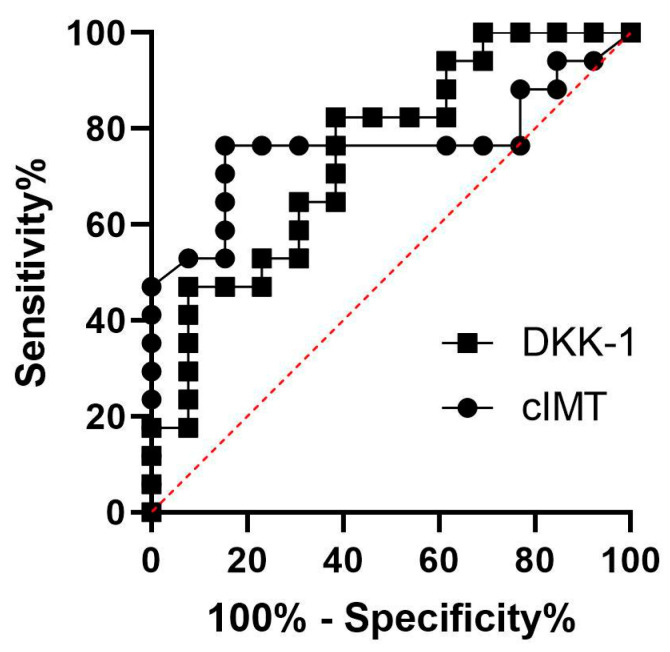
ROC curves for DKK-1 and cIMT.

**Figure 4 ijms-24-14970-f004:**
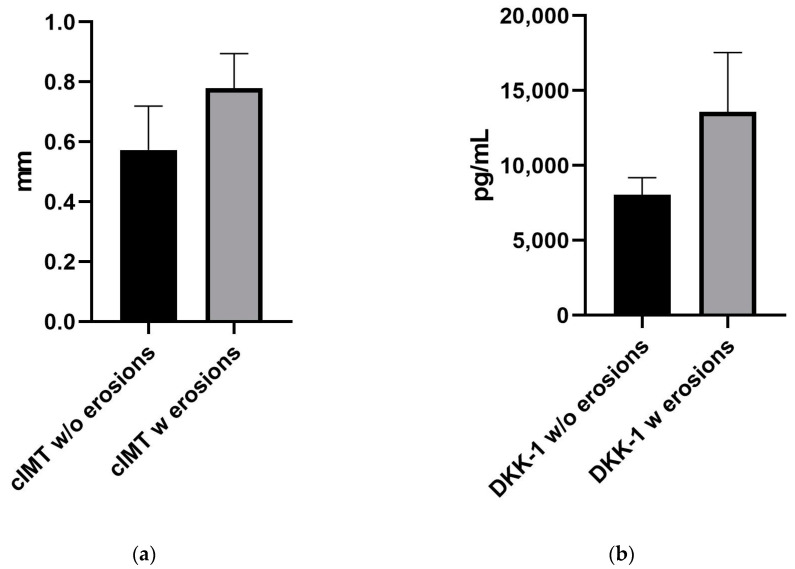
Graphs showing the difference between mean (**a**) cIMT measurement and (**b**) DKK-1 levels for subjects with and without erosions.

**Figure 5 ijms-24-14970-f005:**
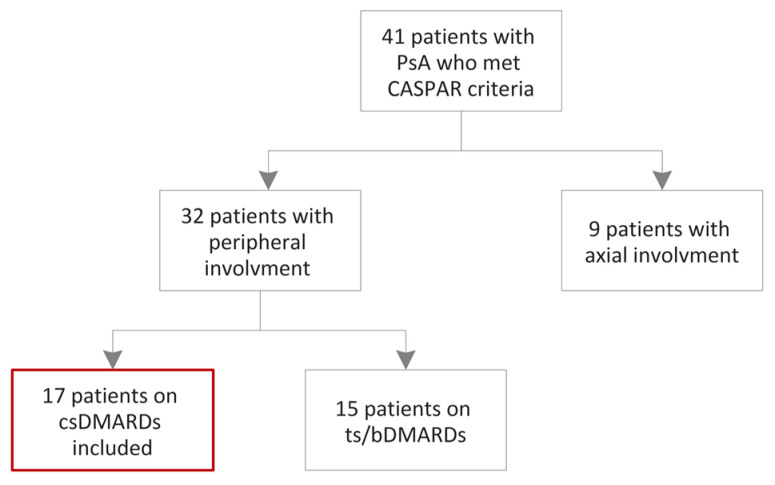
Flowchart of the patients’ inclusion. csDMARDs: conventional synthetic disease-modifying antirheumatic drugs; tsDMARDs: targeted synthetic disease-modifying antirheumatic drugs; bDMARDs: biological disease-modifying antirheumatic drugs.

**Table 1 ijms-24-14970-t001:** Clinical and laboratory variables for the study groups.

	Control Group (*n* = 13)	PsA (*n* = 17)	*p*
Female, *n* (%)	7 (53.85)	9 (52.94)	0.9547
Male, *n* (%)	6 (46.15)	8 (47.06)	
Mean age (years)	50.15 ± 10.11	49.53 ± 9.65	0.8774
Smoker status (current, %)	30.76	35.29	0.9480
Disease duration (years)	-	5.88 ± 4.28	-
DAPSA	-	36.76 ± 12.26	-
PASI	-	17.41 ± 11.12	-
Enthesitis, *n* (presence %)	0 (0)	11 (64.71)	<0.001
Bone erosion, *n* (presence %)	1 (7.69)	9 (52.94)	0.0357
BMI (kg/m^2^)	27.70 ± 4.42	28.89 ± 5.47	0.4621
TC (mg/dL)	193.08 ± 42.17	211.82 ± 53.70	0.4013
HDL (mg/dL)	66.31 ± 12.53	43.18 ± 10.35	0.4681
LDL (mg/dL)	126.77 ± 48.47	168.65 ± 59.25	0.5218
CRP (mg/L)	4.09 ± 2.87	16.90 ± 12.47	<0.001
DKK-1 (pg/mL)	8159.86 ± 2232.04	10,986.2 ± 4063.56	0.0413
cIMT (mm)	0.54 ± 0.09	0.68 ± 0.17	0.0333
Atherosclerotic plaque, *n* (presence %)	3 (23.08)	9 (52.94)	0.5826
SCORE (%)	7.5	13.47	0.0240

**Table 2 ijms-24-14970-t002:** Ultrasonography changes based on MASEI.

	Abnormal Tendon Structure *n* (%)	Thickened Tendon *n* (%)	Erosion *n* (%)	Enthesis Calcification/Enthesophyte *n* (%)	Enthesis PD *n* (%)	Bursitis *n* (%)
TT	1 (5.88)	0 (0.00)	0 (0.00)	1 (5.88)	0 (0.00)	0 (0.00)
QT	3 (17.65)	1 (5.88)	2 (11.76)	2 (11.76)	2 (11.76)	0 (0.00)
PP	1 (5.88)	1 (5.88)	1 (5.88)	1 (5.88)	0 (0.00)	0 (0.00)
DP	3 (17.65)	2 (11.76)	0 (0.00)	1 (5.88)	1 (5.88)	0 (0.00)
AT	6 (35.29)	3 (17.65)	5 (29.41)	5 (29.41)	5 (29.41)	1 (5.88)
PA	2 (11.76)	1 (5.88)	1 (5.88)	1 (5.88)	0 (0.00)	0 (0.00)

TT: tibial tuberosity; QT: quadriceps tendon; PP: proximal patellar tendon; DP: distal patellar tendon; AT: Achilles tendon; PA: plantar aponeurosis. Enthesitis PD was characterized by the detection of Doppler signal within the enthesis, located within less than 2 mm from the cortical bone. Bursitis was identified as the presence of a hypoechoic or anechoic area at the site of a specific bursa.

## Data Availability

Not applicable.

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
