# Peer review of "Dickkopf-Related Protein 1 (DKK-1) as a Possible Link between Bone Erosions and Increased Carotid Intima-Media Thickness in Psoriatic Arthritis: An Ultrasound Study"

_ijms, 2023, doi:10.3390/ijms241914970_

Round 1
Reviewer 1 Report
Dear authors, I read with interest your paper and I think it gives important research insights in the field of Psoriatic Arthritis. Anyway, I have only some concerns about its scientific relevance:
1) As you wrote, the sample size is very small, considering the epidemiology of PsA, and this deeply affects the relevance of your results
2) Included patients were only treated with csDMARDs but I did not understand if treatment with b/tsDMARDs was an exclusion criteria. Nevertheless, disease duration was quite long in the majority of patients and this could modify their cytokines patterns so that we can not assess the utility of DKK-1 in early disease as a markers of CV and erosion risk
3) The definition and the assessment of bone erosion should be better defined
4) Was the correlation between DKK-1 and CRP assessed? Because if there was a strong correlation with this 2 biomarkers, knowing the association between high levels of CRP, erosive disease and cardiovascular risk, DKK-1 could lose its relevance
Author Response
1. As you wrote, the sample size is very small, considering the epidemiology of PsA, and this deeply affects the relevance of your results.
Answer:
Taking into consideration that we are a county hospital, and the duration of the study was only one year, the number of patients with psoriatic arthritis who addressed our clinic was in concordance with the documented statistical data. The justification for this is the small patient population size, due to the fact that the incidence of psoriasis in our country is of 4.99% (10.3390/jpm11060523) and the psoriatic arthritis prevalence is 22.7% in European patients (10.1016/j.jaad.2018.06.027).
2. Included patients were only treated with csDMARDs but I did not understand if treatment with b/tsDMARDs was an exclusion criteria. Nevertheless, disease duration was quite long in the majority of patients and this could modify their cytokines patterns so that we can not assess the utility of DKK-1 in early disease as a markers of CV and erosion risk.
Answer:
Indeed b/tsDMARDS treatment was an exclusion criterion because we selected patients on csDMARDS who had high disease activity and we did this to demonstrate the importance of adequate disease control on cardiovascular comorbidities. Our study highlighted the connection between DKK-1 and cardiovascular disease and erosion risk in patients with a medium duration of the disease. We have planned a future study with patients in early disease stages that could bring more evidence in the foreground.
3. The definition and the assessment of bone erosion should be better defined.
Answer:
We state in the manuscript that the pathological aspects were assessed according to the Outcome Measures in Rheumatology (OMERACT) which include the definition of bone erosions. Also, the only joint imagistic assessment that we used on our patients was the musculoskeletal ultrasonography, thus implying that the bone erosions were identified as such. Indeed, the attached references were related to the classification of enthesitis according to OMERACT, therefore we have added a new reference (10.3899/jrheum.181095) to clarify the definition of bone erosions.
4. Was the correlation between DKK-1 and CRP assessed? Because if there was a strong correlation with this 2 biomarkers, knowing the association between high levels of CRP, erosive disease and cardiovascular risk, DKK-1 could lose its relevance.
Answer:
The correlation that we found between DKK-1 and CRP was not statistically significant, most probably due to the patient population size. The reason why we also consider useful the evaluation of DKK-1 is the fact that psoriatic arthritis, as well as the other pathologies from the spondylarthritis group can be characterized by low CRP levels despite the clinical activity of the disease.
Reviewer 2 Report
In this report, patients with PsA had higher serum levels of Dkk1 than healthy control subjects and were associated with joint erosive disease evaluated by US. Dkk1 was associated with SCORE chart end higher carotid intima-media thickness by the US as well. The authors concluded that these findings would suggest that DKK-1 could be used as an early biomarker for the erosive character of the articular disease and for the assessment of the cardiovascular risk in PsA patients.
Just a couple of corrections to make:
- In the Title (line 4), to change “Artiritis” to “Arthritis”.
- In Table 1, last row, the percentages are shifted. To correct, please.
Author Response
1. Just a couple of corrections to make:
-In the Title (line 4), to change “Artiritis” to “Arthritis”.
-In Table 1, last row, the percentages are shifted. To correct, please.
Answer:
We appreciate the observations and we have modified the suggestions as stated.
Round 2
Reviewer 1 Report
Dear authors, thank you for your explanation.
I am looking forward to read the paper in early disease